# VIVIDFACE: HIGH-QUALITY AND EFFICIENT ONE-STEP DIFFUSION FOR VIDEO FACE ENHANCEMENT

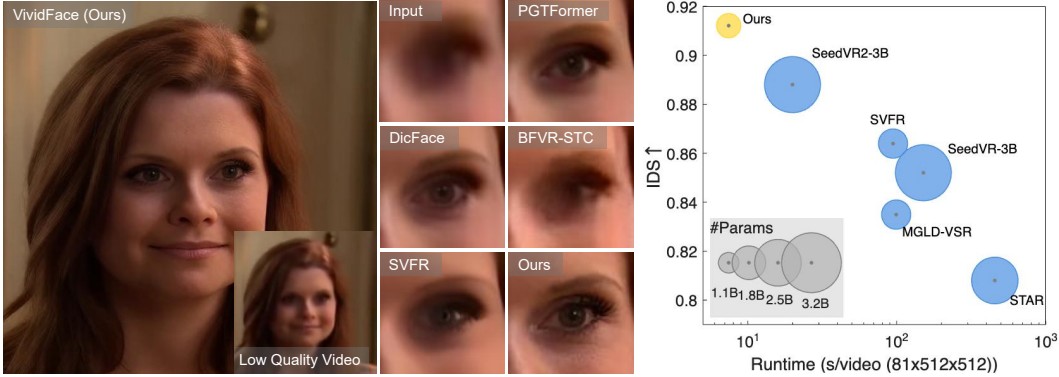

Figure 1: The left side shows a visual comparison between VividFace and existing video face restoration methods, illustrating that VividFace produces highly realistic and visually pleasing human eyes. The right side compares model inference time, parameter count, and IDS performance across different methods. VividFace achieves best performance, fastest speed, and comparable model parameter.

## ABSTRACT

Video Face Enhancement (VFE) seeks to reconstruct high-quality facial regions from degraded video sequences, a capability that underpins numerous applications including video conferencing, film restoration, and surveillance. Despite substantial progress in the field, current methods that primarily rely on video super-resolution and generative frameworks continue to face three fundamental challenges: (1) faithfully modeling intricate facial textures while preserving temporal consistency; (2) restricted model generalization due to the lack of high-quality face video training data; and (3) low efficiency caused by repeated denoising steps during inference. To address these challenges, we propose VividFace, a novel and efficient one-step diffusion framework for video face enhancement. Built upon the pretrained WANX video generation model, our method leverages powerful spatiotemporal priors through a single-step flow matching paradigm, enabling direct mapping from degraded inputs to high-quality outputs with significantly reduced inference time. To further boost efficiency, we propose a Joint Latent-Pixel Face-Focused Training strategy that employs stochastic switching between facial region optimization and global reconstruction, providing explicit supervision in both latent and pixel spaces through a progressive two-stage training process. Additionally, we introduce an MLLM-driven data curation pipeline for automated selection of high-quality video face datasets, enhancing model generalization. Extensive experiments demonstrate that VividFace achieves state-of-the-art results in perceptual quality, identity preservation, and temporal stability, while offering practical resources for the research community.

## 1 INTRODUCTION

Video face enhancement aims to remove degradations from facial videos and improve their details, and has become a fundamental technology for applications such as surveillance systems, entertain-

ment industries, video communication platforms, and digital content creation(Wang et al., 2025d; Zhang & Wu, 2021; Rota et al., 2023). With the continuous development of deep learning, numerous approaches have been proposed, attracting significant attention from researchers(Wang et al., 2025d; Zou et al., 2025; Cao et al., 2025). The key to video face enhancement lies in effectively modeling facial textures and reconstructing realistic facial details, while simultaneously improving the efficiency of video processing.

With the continuous deepening of research, existing methods for video face enhancement can be broadly categorized as follows. General video enhancement methods (Chan et al., 2022a; Liang et al., 2022; Chan et al., 2022b; Yang et al., 2024; Xie et al., 2025; Wang et al., 2025c) lack specialized facial priors and often fail to capture the unique structural characteristics of human faces, resulting in suboptimal restoration of critical facial features. To address these limitations, recent dedicated face video enhancement approaches (Xu et al., 2024b; Feng et al., 2024b; Wang et al., 2025d;e; Chen et al., 2025a) have shown promising results. However, these methods still face several fundamental challenges. First, current methods often struggle to recover sufficient facial details, particularly failing to reconstruct fine-grained textures in key regions such as the eyes, lips, and skin, which results in blurred or unnatural facial appearances, as shown in Figure 1. Second, widely used public datasets like VoxCeleb1 (Nagrani et al., 2020) and VFHQ (Xie et al., 2022) present data quality challenges, as they contain inconsistent degradations including motion blur, poor illumination, and facial occlusions, thereby impeding the effective learning of authentic facial texture structures. Most critically, although diffusion-based approaches have demonstrated strong generative capabilities for high-fidelity reconstruction, they are hindered by significant inference efficiency bottlenecks due to their iterative multi-step sampling processes (Feng et al., 2024b; Wang et al., 2025e; Chen et al., 2025a), making them computationally impractical for real-time or large-scale deployment scenarios where processing speed is crucial.

To address these limitations, we introduce VividFace, an efficient one-step diffusion framework for video face enhancement. Specifically, we build upon the advanced pretrained WANX (Wang et al., 2025a) video generation model to provide strong spatiotemporal priors. By reformulating the traditional multi-step diffusion process into a single-step paradigm using flow matching, our method enables direct mapping from degraded inputs to high-quality outputs, greatly improving inference speed and efficiency. To enhance the restoration of facial details, we propose a Joint Latent-Pixel Face-Focused Training strategy that constructs facial masks aligned with the latent geometry of the VAE encoder, guiding the model to focus on key facial regions during optimization. Furthermore, we develop an MLLM-driven high-quality video filtering pipeline to automatically curate reliable face-centric training data, which helps overcome the data quality issues present in existing training datasets. Extensive experiments on both synthetic and real-world datasets demonstrate the superior performance of our approach compared to existing methods.

The main contributions of this paper are summarized as follows:

- We introduce the first one-step diffusion framework tailored for video face enhancement, achieving a remarkable 12× speedup over SVFR while consistently outperforming existing methods across diverse evaluation metrics on both synthetic and real-world datasets.

- We propose a novel Joint Latent-Pixel Face-Focused Training strategy, which provides explicit facial guidance in both latent and pixel spaces, enabling more targeted optimization of facial regions through a progressive two-stage training process.

- We develop an automated MLLM-driven high-quality video filtering pipeline and present MLLM-Face90, a meticulously curated dataset containing 1,957 high-quality face video clips, empowering the model to learn more authentic facial textures and details.

## 2 RELATED WORK

**Video Face Enhancement.** Face video enhancement aims to recover high-quality facial video from degraded video, and finds application in surveillance, entertainment, and video communication scenarios. Considering the unique facial priors and structural textures, directly applying general Video Super-Resolution (VideoSR) methods (Chan et al., 2022a) to facial videos often fails to achieve satisfactory results. Recent research has focused on dedicated face video enhancement approaches to address unique challenges such as inter-frame flickering, identity drift, and texture inconsistencies.

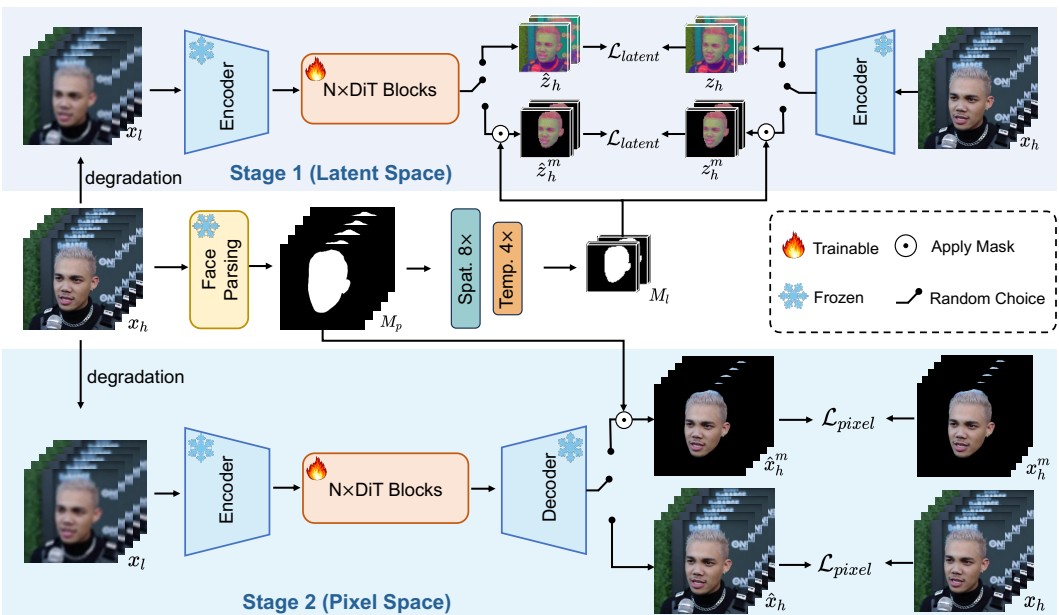

Figure 2: Overview of our proposed VividFace training framework. VividFace is a one-step diffusion method built upon the powerful WANX video model. It adopts a two-stage design that integrates latent and pixel-space optimization, leveraging spatiotemporal priors and stochastic training to simultaneously enhance facial details and overall video quality.

PGTFormer (Xu et al., 2024b) is the first method tailored for video face enhancement, enabling end-to-end enhancement without pre-alignment. KEEP (Feng et al., 2024b) improves temporal consistency by recursively leveraging previously restored frames to guide current frame enhancement. SVFR (Wang et al., 2025e) utilizes generation and motion priors from Stable Video Diffusion for more robust enhancement. DicFace (Chen et al., 2025a) introduces the Dirichlet distribution for continuous codebook combination, offering greater flexibility in representation. Furthermore, current methods are still limited by the low quality of available data. Although existing datasets such as VoxCeleb1 (Nagrani et al., 2020), CelebV-HQ (Zhu et al., 2022), VFHQ (Xie et al., 2022), and FOS (Chen et al., 2024) provide a large amount of facial video data, these datasets often contain degradations such as motion blur, poor lighting, and occlusions, resulting in suboptimal training data quality. These low-quality data samples pose significant challenges for face enhancement methods, making it difficult to efficiently generate realistic facial textures.

**One-step Diffusion.** Diffusion models (Song et al., 2020b; Ho et al., 2020; Song et al., 2020a; Rombach et al., 2022) have achieved impressive visual results in various tasks, thanks to their ability to generate high-fidelity and realistic frames. However, their multi-step inference process leads to high computational cost and slow generation, especially for video data where efficiency is crucial. Recently, one-step diffusion methods have been proposed to accelerate generation, and have shown promising results in both image (Li et al., 2024; Wu et al., 2024; Wang et al., 2024a; Dong et al., 2025) and video super-resolution (Chen et al., 2025b; Wang et al., 2025b; Sun et al., 2025; Liu et al., 2025). Despite these advances, the application of one-step diffusion remains largely unexplored for blind face video enhancement, leaving a gap in this important area.

## 3 METHODOLOGY

In this section, we present VividFace, a novel framework for face video enhancement that achieves strong visual quality and performance with low computational cost. Our approach consists of three main components: a one-step flow matching framework based on the pretrained WANX model, as shown in Sec. 3.1; a Joint Latent-Pixel Face-Focused Training strategy that directs attention to key facial regions, as shown in Sec. 3.2; and an MLLM-driven high-quality video filtering pipeline for building high-quality face-centric training datasets, as shown in Sec. 3.3.

### 3.1 ONE STEP FLOW MATCHING FOR VIDEO DIFFUSION MODELS

We propose VividFace, a one-step face video enhancement network, leveraging the powerful pre-trained text-to-video generation model WANX. WANX adopts an encoder–DiT–decoder architecture to model video generation in the latent space via flow matching, and is pretrained on large-scale real-world video datasets, enabling it to acquire robust generative priors for video enhancement. By capitalizing on generative priors, our model is able to effectively handle diverse and complex degradation conditions. Furthermore, to reduce inference time, we reformulate the multi-step flow matching into a single-step paradigm, enabling direct transformation from degraded video inputs to high-quality video outputs, achieving fast speed.

The overall architecture of our one-step model is illustrated in Fig. 2. Following Rectified Flows (Esser et al., 2024), given a low-quality face video $x_l$, we first encode it into a latent representation $z_l$ using the VAE encoder $\mathcal{E}$. We designate $z_l$ as the flow starting point, establishing a continuous flow trajectory between degraded input $z_l$ and high-quality target $z_h$, as follows:

$$z_t = (1 - t)z_l + tz_h, \quad t \in [0, 1],\tag{1}$$

where $t$ represents the time step. The ground truth velocity field is as follow:

$$v_t = \frac{dz_t}{dt} = z_h - z_l.\tag{2}$$

The DiT model is trained to predict this velocity field $v_t$ through a single denoising step, and obtain generated high-quality latent $\hat{z}_h$, as follows:

$$\hat{z}_h = z_l + v_\theta(z_l, t^*, c_{txt}),\tag{3}$$

where $v_\theta$ represents the velocity predicted by the DiT model, and $c_{txt}$ is the text embedding. To facilitate efficient training of the DiT model, we employ the VAE to pre-extract the latent representations of both low-quality and high-quality videos. Additionally, we use empty prompts to eliminate the computational overhead associated with text captions. We empirically set $t^* = 400$ in the original discrete timestep scale (corresponding to $t \approx 0.4$ in the continuous scale), which balances structural preservation and detail enhancement based on the observation that the low-resolution input already contains sufficient structural information. Finally, the enhanced latent representation $\hat{z}_h$ is decoded through the VAE decoder $\mathcal{D}$ to produce the output video $\hat{x}_h$ as the RGB enhancement result.

### 3.2 JOINT LATENT-PIXEL FACE-FOCUSED TRAINING

Standard diffusion models treat all spatial regions equally, often underutilizing the capacity for critical facial details. Motivated by key insights that facial regions contain the most critical information for perceptual quality in face enhancement tasks, we introduce a Joint Latent-Pixel Face-Focused Training strategy that explicitly guides the model to concentrate on facial regions. Our approach provides mask-guided supervision in both latent and pixel spaces by constructing facial masks aligned with the corresponding spatiotemporal geometry of each space.

The core of our approach lies in constructing facial masks aligned with the spatiotemporal geometry of the VAE encoder, enabling targeted supervision in latent space. Given an input video $x_l \in \mathbb{R}^{(1+T) \times H \times W \times 3}$ encoded into latent representation $z_l = \mathcal{E}(x_l) \in \mathbb{R}^{C \times T' \times H' \times W'}$ (where $C = 16$, $H' = H/8$, $W' = W/8$, and $T' = 1 + T/4$), we need to create corresponding facial masks at the latent resolution. Instead of the computationally expensive alternative of encoding masks through the VAE, we adopt an efficient geometric alignment strategy. As illustrated in Fig. 2 (middle), we first extract per-frame binary facial masks from the ground truth video $x_h$ using a face parsing model (Yu et al., 2018), yielding $M_p \in \{0, 1\}^{(1+T) \times H \times W}$. To align these masks with the latent geometry, we perform spatial downsampling by a factor of $8 \times 8$ via nearest-neighbor interpolation: $\widetilde{M} = \mathcal{D}_s(M_p) \in [0, 1]^{(1+T) \times H' \times W'}$. The temporal alignment follows the encoder's compression scheme. For latent temporal indices $i = 0, \ldots, T' - 1$, we set $\widehat{M}^{(0)} = \widetilde{M}^{(0)}$ for the first frame, while subsequent frames aggregate every four consecutive frames through element-wise maximum operation: $\widehat{M}^{(i)} = \max_{j=1}^{4} \widetilde{M}^{(4(i-1)+j)}, \quad i = 1, \ldots, T' - 1$. This produces a spatiotemporally aligned facial mask $\widehat{M} \in [0, 1]^{T' \times H' \times W'}$, which is then replicated along the channel dimension to obtain $M_l \in [0, 1]^{C \times T' \times H' \times W'}$ that matches the latent representation $z_l$.

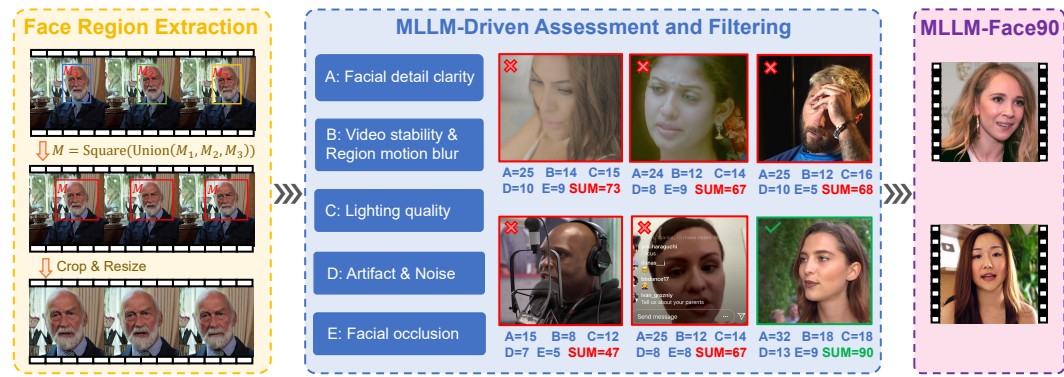

Figure 3: Pipeline of the proposed MLLM-driven high-quality face video filtering. First, face regions are extracted and cropped to facilitate the model's focus on facial features. Next, a meticulously designed set of visual quality assessment prompts is utilized to evaluate each video from multiple quality perspectives using the powerful Qwen2.5-VL.

With the facial masks properly constructed, we integrate them into both latent and pixel spaces to provide explicit supervision during training. As illustrated in Fig. 2, we employ a stochastic training strategy that dynamically switches between face-focused and global reconstruction objectives. Formally, for a given representation space with ground-truth $y$, prediction $\hat{y}$, and mask $M$, the loss function is defined as:

$$\mathcal{L}(y, \hat{y}, M) = b \cdot \left\| M \odot \hat{y} - M \odot y \right\|_2^2 + (1 - b) \cdot \left\| \hat{y} - y \right\|_2^2, \quad b \sim \text{Bernoulli}(p), \qquad (4)$$

where $\mathcal{L}_{\text{latent}}$ corresponds to $(y, \hat{y}, M) = (z_h, \hat{z}_h, M_l)$ in latent space and $\mathcal{L}_{\text{pixel}}$ corresponds to $(y, \hat{y}, M) = (x_h, \hat{x}_h, M_p)$ in pixel space.

To effectively leverage this dual-space supervision, we design a progressive two-stage training strategy. Specifically, in the first stage, the model learns to fit the one-step flow trajectory by optimizing $\mathcal{L}_{\text{latent}}$, establishing robust spatiotemporal priors and coarse facial structure recovery in the latent space. In the second stage, fine-tuning is performed in pixel space using $\mathcal{L}_{\text{pixel}}$ augmented with perceptual supervision: $\mathcal{L}_{\text{pixel}} + \lambda \mathcal{L}_{\text{DISTS}}$, where $\mathcal{L}_{\text{DISTS}}$ (Ding et al., 2020) enhances perceptual quality and fine-grained detail generation. This progressive approach enables the model to first establish a solid foundation in latent space before refining pixel-level facial textures, resulting in superior restoration quality and temporal consistency.

## 3.3 MLLM-Driven High-Quality Video Filtering

High-quality face video data is indispensable for the training of video face restoration models. However, existing face-centric datasets such as VFHQ exhibit two major limitations: (1) The facial region typically occupies only a small fraction of each frame, causing models to overfit to background restoration and making it difficult to sufficiently learn detailed facial texture structures; (2) The datasets contain a large proportion of low-quality samples with severe degradations such as motion blur, poor illumination, and occlusions, which leads to outputs with artifacts and unrealistic textures due to the lack of authentic facial details in the training data. These shortcomings hinder face restoration methods from generating high-quality facial textures and details. To address these challenges, we propose a novel MLLM-driven high-quality video filtering pipeline, termed **MLLM-Face90**, which provides a curated, face-centric dataset, as shown in Fig. 3.

**Face Region Extraction.** We employ a state-of-the-art face parsing model (Yu et al., 2018) to precisely localize facial regions in each video frame. This targeted extraction ensures that subsequent quality assessment and filtering are exclusively performed on pertinent facial areas, thereby minimizing interference from background or non-facial elements. Such isolation enhances the reliability and relevance of downstream evaluations for face restoration applications.

**MLLM-Driven Quality Assessment and Filtering.** In the second step, we establish a rigorous, automated quality assessment pipeline leveraging a Qwen2.5-VL (Bai et al., 2025). The MLLM is guided by a meticulously designed prompt that instructs it to evaluate each candidate video under strict, multi-dimensional criteria specifically tailored for face restoration tasks. The evaluation

covers five key dimensions: facial detail clarity, video stability and regional motion blur, lighting quality, artifact and noise level, and facial occlusion. For each dimension, the MLLM assigns a score according to well-defined rubrics, with further bonus and penalty points reflecting exceptional strengths or notable deficiencies. The prompt explicitly directs the MLLM to focus on critical facial regions—including the eyes, mouth/lips, teeth, and nose—and to penalize any motion blur or degradation in these areas. It emphasizes the necessity for natural or studio-quality lighting, the absence of compression artifacts and digital noise, and the full visibility of all key facial features. Through the implementation of a stringent multi-dimensional evaluation protocol, only those videos achieving a superior quality assessment score (typically exceeding 90 out of 100) are retained as high-quality face video samples and incorporated into our benchmark dataset, **MLLM-Face90**. Subsequent fine-tuning on MLLM-Face90 yields notable performance gains, as demonstrated in Section 4.3.

## 4 EXPERIMENTS

### 4.1 IMPLEMENTATION DETAILS

**Training Details.** We adopt a coarse-to-fine training strategy. Specifically, we first randomly sample 3,000 video clips from VFHQ (Xie et al., 2022) for coarse training of VividFace. Subsequently, fine-tuning is performed using MLLM-Face90, a carefully curated dataset comprising 1,957 high-quality clips. Furthermore, we follow previous method (Feng et al., 2024b) to synthesize low-quality data: $y = [(x \circledast k_\sigma) \downarrow_r + n_\delta]_{\text{FFMPEG}_{\text{crf}}}$, where $y$ and $x$ denote the low-quality (LQ) and high-quality (HQ) videos, respectively. $\circledast$ represents the convolution operation, $k_\sigma$ and $n_\delta$ are the Gaussian blur kernel and Gaussian noise, and $\downarrow_r$ indicates $r\times$ downsampling. During preprocessing, $\sigma$, $r$, $\delta$, and crf are randomly sampled from $[0.1, 10]$, $[1, 4]$, $[0, 10]$, and $[18, 25]$, respectively. The hyperparameters $\lambda$ and $p$ are empirically set to 0.1 and 0.5, respectively. The frames of the two training stages are $81 \times 512 \times 512$ and $13 \times 512 \times 512$, respectively. All experiments are conducted on eight NVIDIA A100 GPUs, with a batch size of 32 and a learning rate of $1 \times 10^{-4}$ for a total of 32,000 iterations.

**Evaluation.** To comprehensively validate the performance of our method, we conduct experiments on both synthetic and real-world benchmarks. For synthetic evaluation, following previous works, we employ the official VFHQ-test dataset (Xie et al., 2022) with the aforementioned degradation model, which consists of 50 high-quality (HQ) video clips. Additionally, to assess performance in real-world scenarios, we adopt the RFV-LQ dataset (Wang et al., 2024b), following prior protocols. RFV-LQ contains 329 low-quality face videos meticulously curated from diverse real-world sources, including old talk shows, TV series, and movies, providing a robust testbed for evaluating the method's robustness across various real-world conditions.

**Metrics.** To facilitate a comprehensive and rigorous evaluation, we employ a diverse set of video quality assessment metrics spanning multiple dimensions of model performance. Specifically, we assess results from three key perspectives: Quality and Fidelity, Pose Consistency, and Temporal Consistency. For Quality and Fidelity, we use six representative metrics: PSNR, SSIM (Wang et al., 2004), and LPIPS (Zhang et al., 2018) (reference-based), as well as NIQE (Mittal et al., 2012), MUSIQ (Ke et al., 2021), and CLIP-IQA (Wang et al., 2023) (no-reference). To measure Pose Consistency, we adopt IDS, AKD, and FaceCons (Feng et al., 2024a), as well as TLME (Xu et al., 2024a). We multiplie AKD and TLME by 1000, denoted as AKD* and TLME*, respectively. For Temporal Consistency, we employ FasterVQA (Wu et al., 2023) and FVD (Unterthiner et al., 2019). This comprehensive selection of metrics enables a robust and systematic evaluation of video face enhancement performance.

**Compared Methods.** We evaluate our approach against three distinct categories of classic and representative compared methods. First, we select widely-used Video Super-Resolution (VSR) models, including BasicVSR++ (Chan et al., 2022a), RealBasicVSR (Chan et al., 2022b), MGLD-VSR (Yang et al., 2024), STAR (Xie et al., 2025), SeedVR (Wang et al., 2025c), and SeedVR2 (Wang et al., 2025b), which enhance overall video quality but lack specialized facial restoration. Second, we include established Face Image-based Restoration (FIR) models, such as CodeFormer (Zhou et al., 2022) and DifFace (Yue & Loy, 2024), which restore facial details independently for each frame. Third, we compare with state-of-the-art Face Video-based Restoration (FVR) models that leverage both spatial and temporal information, including PGTFormer (Xu et al., 2024b), BFVR-STC (Wang et al., 2025d), KEEP (Feng et al., 2024b), SVFR (Wang et al., 2025e),

Table 1: Quantitative comparison on the VFHQ-test dataset. The best and second-best methods are highlighted in red and blue respectively. VividFace achieves superior performance, outperforming existing methods across **all metrics and benchmarks**.

| Method | | Quality and Fidelity | | | Pose Consistency | | | Temporal Consistency | |
|---|---|---|---|---|---|---|---|---|---|
| | | PSNR↑ | SSIM↑ | LPIPS↓ | IDS↑ | AKD*↓ | FaceCons↑ | FasterVQA↑ | FVD↓ |
| VSR | BasicVSR++ | 25.64 | 0.7860 | 0.3902 | 0.7796 | 9.2778 | 0.7475 | 0.2108 | 1073.70 |
| | RealBasicVSR | 26.82 | 0.7793 | 0.2656 | 0.8046 | 6.3262 | 0.7442 | 0.8086 | 308.90 |
| | MGLD-VSR | 27.37 | 0.8111 | 0.2151 | 0.8350 | 4.6829 | 0.7297 | 0.7977 | 285.30 |
| | STAR | 24.66 | 0.7729 | 0.3393 | 0.8077 | 6.6215 | 0.7605 | 0.6643 | 464.51 |
| | SeedVR-3B | 27.04 | 0.7860 | 0.2271 | 0.8523 | 4.4892 | 0.7905 | 0.8025 | 126.14 |
| | SeedVR2-3B | 27.75 | 0.8420 | 0.1538 | 0.8887 | 3.9975 | 0.8013 | 0.8194 | 116.56 |
| FVR | PGTFormer | 28.78 | 0.8460 | 0.1837 | 0.8612 | 4.3572 | 0.7298 | 0.8484 | 197.02 |
| | BFVR-STC | 24.37 | 0.7858 | 0.3383 | 0.7793 | 8.2443 | 0.7179 | 0.4594 | 700.20 |
| | KEEP | 27.50 | 0.8152 | 0.2376 | 0.7950 | 4.5966 | 0.7529 | 0.7986 | 388.60 |
| | SVFR | 28.09 | 0.8304 | 0.1578 | 0.8641 | 4.0932 | 0.8025 | 0.8404 | 103.32 |
| | DicFace | 28.25 | 0.8313 | 0.2424 | 0.8854 | 3.9682 | 0.7634 | 0.7207 | 340.76 |
| **VividFace (Ours)** | | 30.03 | 0.8534 | 0.1112 | 0.9128 | 3.5319 | 0.8111 | 0.8855 | 79.14 |

Table 2: Quantitative comparison on the **real-world** RFV-LQ dataset. The best and second-best methods are highlighted in red and blue respectively. Our method consistently demonstrates superior performance, highlighting its **robustness** in real-world scenarios.

| Method | | Quality and Fidelity | | | Pose Consistency | | Temporal Consistency |
|---|---|---|---|---|---|---|---|
| | | NIQE↓ | MUSIQ↑ | CLIP-IQA↑ | TLME*↓ | FaceCons↑ | FasterVQA↑ |
| VSR | BasicVSR++ | 6.2983 | 30.8569 | 0.2005 | 7.3107 | 0.7207 | 0.2651 |
| | RealBasicVSR | 5.0402 | 63.1429 | 0.5407 | 6.5338 | 0.7392 | 0.7305 |
| | MGLD-VSR | 5.9269 | 62.7775 | 0.5593 | 6.2764 | 0.7359 | 0.7630 |
| | STAR | 5.5227 | 64.4846 | 0.5416 | 6.4900 | 0.7222 | 0.7657 |
| | SeedVR-3B | 5.3900 | 52.8371 | 0.4759 | 6.6668 | 0.7430 | 0.6649 |
| | SeedVR2-3B | 6.6548 | 54.4296 | 0.4051 | 6.5437 | 0.7593 | 0.6059 |
| FVR | PGTFormer | 6.7676 | 59.5214 | 0.4709 | 6.2961 | 0.7151 | 0.7744 |
| | BFVR-STC | 6.8477 | 45.5984 | 0.3372 | 6.7475 | 0.6928 | 0.5149 |
| | KEEP | 6.2016 | 60.9558 | 0.5054 | 6.1623 | 0.7392 | 0.7255 |
| | SVFR | 7.0772 | 54.2877 | 0.3907 | 6.1478 | 0.7527 | 0.6286 |
| | DicFace | 6.8448 | 53.9808 | 0.4525 | 6.2385 | 0.7413 | 0.6185 |
| **VividFace (Ours)** | | 5.1987 | 64.4911 | 0.5678 | 6.0064 | 0.7665 | 0.8227 |

and DicFace (Chen et al., 2025a). For all experiments, we use the same degradation settings as described in our methodology. Each baseline is implemented following the official configurations from their original papers. Several methods, including CodeFormer, DiffFace, KEEP, and DicFace, are restricted to processing aligned facial regions. Accordingly, we adopt the standard KEEP pipeline: the background is first enhanced using Real-ESRGAN (Wang et al., 2021), after which the processed facial regions are seamlessly composited back into the original frames.

## 4.2 COMPARISON WITH STATE-OF-THE-ART

**Quantitative Results.** We present a comprehensive performance comparison on the VFHQ-test benchmark in Table 1 and the real-world RFV-LQ benchmark in Table 2. Our VividFace achieves outstanding performance, surpassing both VSR and FVR methods across all evaluated metrics. Specifically, in Table 1, VividFace significantly outperforms previous methods in terms of PSNR, LPIPS, and FasterVQA, achieving consistent superiority across multiple metrics. In Table 2, various no-reference video quality assessment metrics in real-world scenarios further validate the robustness and superior visual quality of VividFace.

**Qualitative Results.** Comprehensive qualitative results on the VFHQ-test and real-world RFV-LQ datasets are presented in Figure 4 and Figure 5. Existing methods often produce blurred or over-

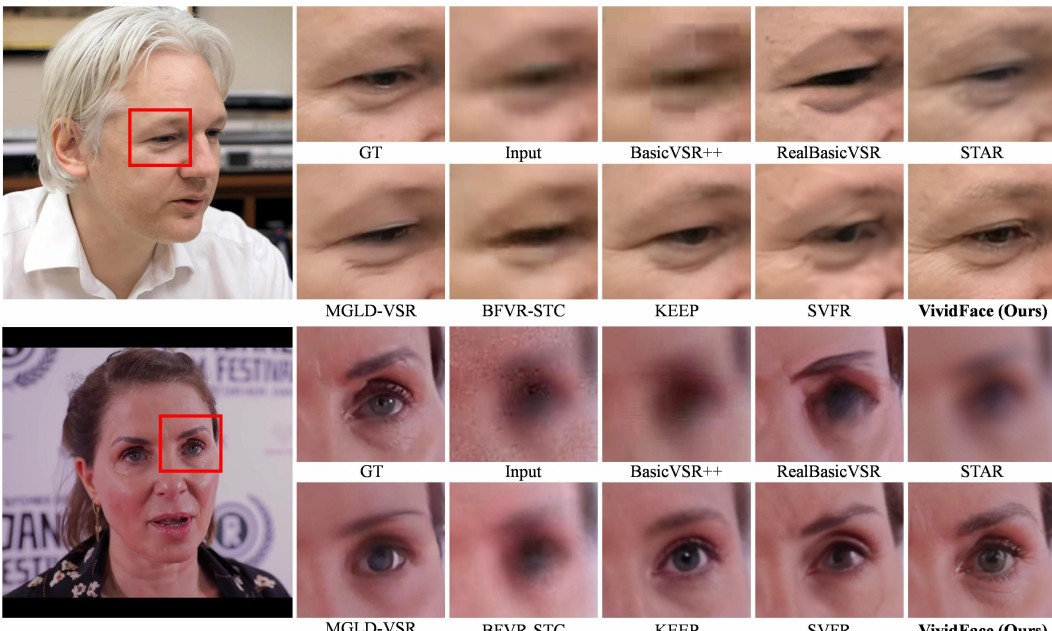

Figure 4: Visual comparison with existing methods on VFHQ-test. VividFace exhibits more realistic and visually pleasing facial details, and produces results that are closer to the ground truth.

Table 3: Running time comparison on 81-frame $512 \times 512$ videos for various methods. VividFace achieves the fastest inference while maintaining superior visual quality and identity preservation.

| Method | Step | Time (s)↓ | LPIPS↓ | IDS↑ |
|---|---|---|---|---|
| MGLD | 50 | 98.73 | 0.215 | 0.835 |
| STAR | 15 | 456.83 | 0.339 | 0.808 |
| SeedVR | 50 | 151.43 | 0.227 | 0.852 |
| SeedVR2 | 1 | 19.98 | 0.153 | 0.888 |
| SVFR | 30 | 94.60 | 0.157 | 0.864 |
| **VividFace (Ours)** | 1 | **7.43** | **0.113** | **0.912** |

smoothed details and fail to reconstruct realistic, identity-consistent structures across key regions such as the eyes, eyebrows, and mouth. In contrast, VividFace demonstrates superior capability in recovering more realistic and visually pleasing results: it faithfully restores fine-grained features while preserving surrounding facial structures, maintaining consistency with the ground-truth identity, and avoiding the over-smoothing artifacts observed in competing approaches. Additional qualitative results are provided in the **supplementary material**.

**Running Time Comparisons.** Table 3 presents a side-by-side comparison of inference speed and model performance across several leading methods, all evaluated on an identical hardware setup (single 80GB A100 GPU, 81-frame videos at $512 \times 512$ resolution). Notably, VividFace completes inference in just **7.43 seconds**, making it approximately **12× faster** than SVFR and **2.7× faster** than SeedVR2-3B, the latter being the closest one-step competitor. Crucially, VividFace achieves this speed without compromising on perceptual quality or identity preservation, as evidenced by its superior LPIPS and IDS scores. These results highlight VividFace's clear advantage in both efficiency and output fidelity.

### 4.3 ABLATION STUDIES

**Effect of Stochastic Face-Focused Training Probability $p$.** We investigate the impact of the stochastic training probability $p$ in Eq. 4, which controls the balance between face-focused and global reconstruction objectives. As shown in Table 4, we evaluate three different values: 0 (pure global training), 0.5 (balanced stochastic training), and 1 (pure face-focused training). The results demonstrate that $p = 0.5$ achieves optimal performance. Pure global training leads to suboptimal

Table 4: Effect of face-focused training probability $p$.

| $p$ | LPIPS↓ | IDS↑ |
|---|---|---|
| 0 | 0.1229 | 0.9073 |
| **0.5** | **0.1112** | **0.9128** |
| 1 | 0.1309 | 0.9025 |

Table 5: Ablation study on face-focused training strategy.

| Latent | Pixel | LPIPS↓ | IDS↑ |
|---|---|---|---|
| × | × | 0.1229 | 0.9073 |
| ✓ | × | 0.1261 | 0.9050 |
| ✓ | ✓ | **0.1112** | **0.9128** |

Table 6: Ablation study on model performance with our proposed MLLM-Face90 dataset.

| Dataset | LPIPS↓ | IDS↑ |
|---|---|---|
| VFHQ-3K | 0.1285 | 0.9046 |
| **+Ours** | **0.1112** | **0.9128** |

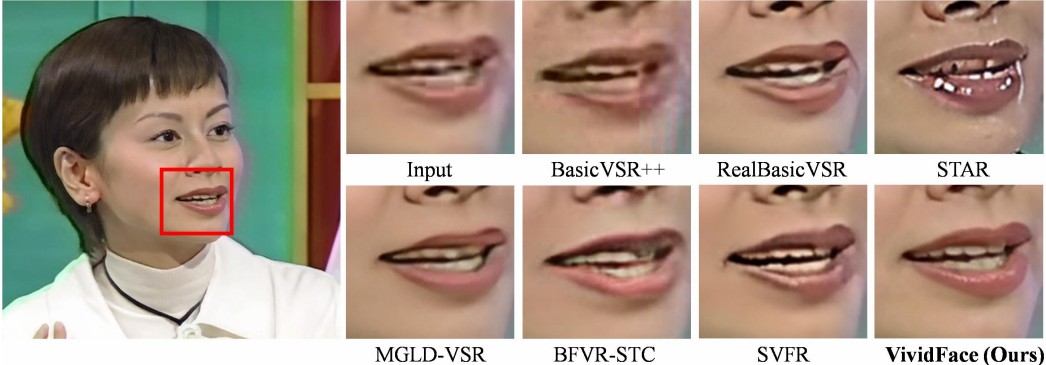

Figure 5: Qualitative comparison on real-world RFV-LQ dataset. The results highlight VividFace's capability to address complex real-world degradations.

facial detail recovery, while pure face-focused training compromises overall video quality due to insufficient global context learning. The balanced approach effectively combines both objectives.

**Effectiveness of Joint Latent-Pixel Training Strategy.** We validate our Joint Latent-Pixel Face-Focused Training strategy by progressively adding training components. Table 5 presents three configurations: baseline without face-focused training, latent space only, and joint training in both spaces. Adding latent space training alone slightly degrades performance, suggesting that latent-only optimization is insufficient for fine-grained facial detail recovery. However, the complete joint training strategy significantly improves both metrics, demonstrating that multi-space optimization is crucial for optimal facial enhancement quality.

**Impact of MLLM-Face90 High-Quality Dataset.** We evaluate the contribution of our MLLM-Face90 dataset by comparing models trained on different data configurations. Table 6 compares using only VFHQ-3K versus incorporating our curated dataset. The incorporation of MLLM-Face90 leads to substantial performance improvements across both metrics, validating the effectiveness of our MLLM-driven data curation pipeline in providing high-quality face-centric training data for learning authentic facial textures.

## 5 CONCLUSION

In this work, we present VividFace, a novel and efficient one-step diffusion framework for video face enhancement. By leveraging the powerful WANX video model as the backbone, our approach benefits from robust spatio-temporal representations, enabling more accurate and consistent restoration of facial details across frames. The integration of our Joint Latent-Pixel Face-Focused Training strategy with stochastic switching between facial and global optimization objectives within a single-step flow matching paradigm significantly accelerates inference while preserving high perceptual quality. Furthermore, our MLLM-driven data curation pipeline facilitates automated construction of high-quality face video datasets, further enhancing model generalization and robustness. Extensive experiments demonstrate that VividFace achieves state-of-the-art performance in perceptual quality, identity preservation, and temporal stability. We believe our work demonstrates the potential of combining advanced video backbones with efficient generative frameworks for high-fidelity video face restoration and provides useful resources for future research.

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

# APPENDIX

## A  MORE VISUAL AND PERFORMANCE COMPARISON

**Visualization of temporal consistency comparison.** Temporal consistency is a crucial aspect of video enhancement. Therefore, we further compare the temporal consistency performance of existing methods, as shown in Figure 6. Specifically, we select the region marked by the red line and display its continuous representations across different frames. It can be observed that Vivid-Face demonstrates superior temporal consistency with significantly reduced jitter compared to other approaches, and its results are much closer to the ground truth.

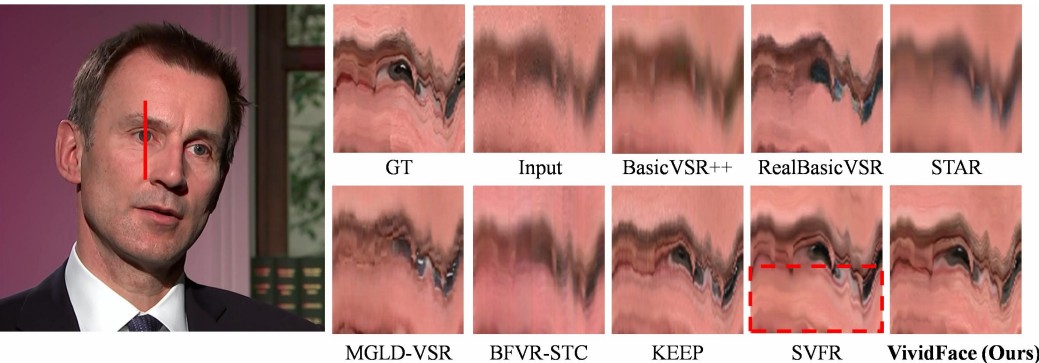

Figure 6: Visualization of temporal consistency comparison. VividFace achieves temporal results that are closer to the ground truth, verifying its stronger temporal consistency.

**More performance comparisons with face restoration methods.** We also compare our method with several face image restoration (FIR) methods on the VFHQ-test dataset. As shown in Table 7, FIR methods perform significantly worse than ours in terms of pose consistency and temporal consistency. This demonstrates the exceptional capability of our approach in face video restoration.

**More qualitative comparisons with face restoration methods.** An extra qualitative comparison on the VFHQ-test dataset is shown in Figure 7. Our VividFace faithfully recovers skin details such as wrinkles and texture around the eye region, further demonstrating its capability for realistic facial enhancement.

Table 7: Comparisons of face image restoration (FIR) methods on the VFHQ-test dataset.

| Method | | Quality and Fidelity | | | Pose Consistency | | | Temporal Consistency | |
| --- | --- | --- | --- | --- | --- | --- | --- | --- | --- |
| | | PSNR↑ | SSIM↑ | LPIPS↓ | IDS↑ | AKD*↓ | FaceCons↑ | FasterVQA↑ | FVD↓ |
| FIR | CodeFormer (Zhou et al., 2022) | 27.27 | 0.8023 | 0.2391 | 0.7719 | 5.5370 | 0.7156 | 0.8625 | 331.99 |
| | DifFace (Yue & Loy, 2024) | 26.73 | 0.7919 | 0.2538 | 0.5998 | 7.6810 | 0.4544 | 0.8278 | 904.14 |
| **VividFace (Ours)** | | 30.03 | 0.8534 | 0.1112 | 0.9128 | 3.5319 | 0.8111 | 0.8855 | 79.14 |

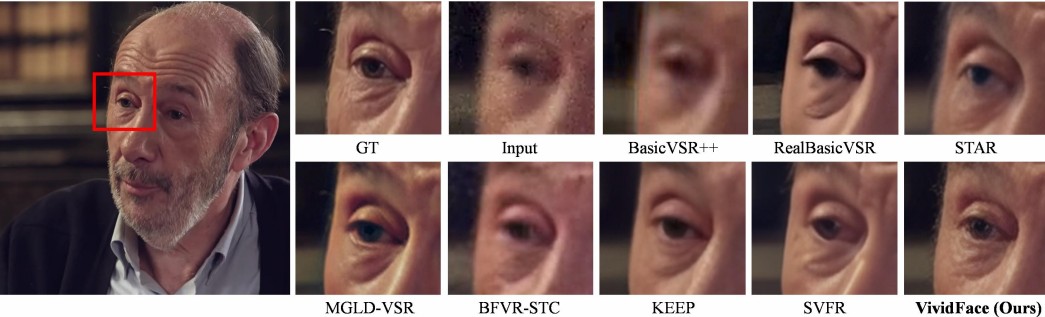

Figure 7: More qualitative comparison on VFHQ-test dataset. VividFace effectively recovers subtle skin details and enhances perceptual quality.

## B  MORE DETAILS OF MLLM PROMPT

In this section, we present a detailed MLLM prompt designed for multi-dimensional video quality assessment. The prompt covers several key criteria, including **facial detail clarity**, **video stability and motion blur**, **lighting quality**, **artifact and noise level**, and **facial occlusion**. Additionally, it requires separate evaluation of critical facial regions such as the eyes, mouth, teeth, and nose. The scoring system incorporates both bonus and penalty mechanisms to ensure a comprehensive and rigorous selection of premium training data. The full prompt structure, showcasing these carefully crafted dimensions, is illustrated in Listing 1.

Listing 1: Premium Face Video Quality Evaluation Prompt

```
You are an expert face video quality inspector specializing in premium face restoration
    training data. Evaluate this video with STRICT criteria to identify only the highest
    quality samples suitable for premium model training.

**Evaluation Criteria (Total: 100 points) - STRICT GRADING:**

1. **Facial Detail Clarity (35 points)**
  - 0-12: Severely degraded, facial features barely distinguishable
  - 13-21: Moderate quality, basic features visible but lacking fine details
  - 22-28: Good quality with visible skin texture and facial features, BUT penalize if key
      regions (eyes, mouth, teeth) show motion blur
  - 29-32: Excellent clarity with clear pores, fine lines, and detailed texture across ALL
      facial regions
  - 33-35: Perfect clarity with crisp micro-details (individual eyelashes, teeth edges, lip
      texture clearly visible)

2. **Video Stability & Regional Motion Blur (20 points)**
  - 0-6: Severe motion blur or instability affecting entire face
  - 7-11: Noticeable camera shake OR significant motion blur in key facial regions (eyes,
      mouth, teeth)
  - 12-15: Minor overall stability issues, but critical facial features remain sharp
  - 16-18: Very stable with minimal motion blur, all key facial regions clear
  - 19-20: Perfect stability across all frames, no motion blur in any facial region

3. **Lighting Quality (20 points)**
  - 0-6: Extreme lighting conditions that obscure facial features
  - 7-11: Acceptable lighting with noticeable issues (uneven shadows, slight over/under
      exposure)
  - 12-15: Good lighting with minor imperfections
  - 16-18: Excellent natural lighting with proper facial modeling
  - 19-20: Perfect studio-quality lighting with optimal facial structure revelation

4. **Artifact & Noise Level (15 points)**
```

```
   – 0-4: Heavy compression artifacts, noise, or digital distortions
   – 5-7: Noticeable artifacts that affect facial details
   – 8-10: Minor artifacts present but don't significantly impact quality
   – 11-13: Minimal artifacts, high video quality
   – 14-15: No visible artifacts, pristine video quality

5. **Facial Occlusion (10 points)**
   – 0-2: Significant occlusion (>25% of face covered by objects, hands)
   – 3-4: Moderate occlusion (10-25% covered)
   – 5-6: Minor occlusion (5-10% covered), most facial features visible
   – 7-8: Minimal occlusion (<5% covered), all key facial features clearly visible
   – 9-10: No occlusion, complete facial visibility

**Critical Facial Regions Check:**
- Eyes: Must be sharp with visible iris details, eyelashes clearly defined
- Mouth/Lips: Lip texture and edges must be crisp, no blur during speech
- Teeth: Individual teeth edges must be clearly visible when shown
- Nose: Nostril details and nose bridge must be sharp

**STRICT QUALITY THRESHOLDS:**
- Premium Training Data: Score  85 (Top 10-15% of videos)
- High-Quality Training Data: Score  80 (Top 20-25% of videos)
- Standard Training Data: Score  75 (Top 40% of videos)
- Below Standard: Score < 75 (Consider discarding for premium training)

**Additional Quality Factors (Bonus/Penalty):**
- **Bonus (+2 points)**: Exceptional skin texture detail visible throughout video
- **Bonus (+1 point)**: Perfect color reproduction and white balance
- **Penalty (-3 points)**: Motion blur detected in ANY key facial region (eyes, mouth, teeth)
    even if brief
- **Penalty (-2 points)**: Any visible digital noise or grain
- **Penalty (-3 points)**: Unnatural skin smoothing or beauty filter effects
- **Penalty (-2 points)**: Inconsistent sharpness between frames (some frames sharp, others
    blurry)

**EVALUATION PROCESS - FOLLOW THESE STEPS:**

1. First, evaluate each criteria and assign a specific score:
   – Clarity: ___/35
   – Stability: ___/20
   – Lighting: ___/20
   – Artifacts: ___/15
   – Occlusion: ___/10

2. Calculate the base score by adding the five scores above:
   Base Score = Clarity + Stability + Lighting + Artifacts + Occlusion = ___

3. Apply bonus/penalty adjustments:
   – List each bonus/penalty with the reason
   – Calculate adjustment total: ___
   – Final Score = Base Score + Adjustment = ___

4. Determine quality tier based on final score

**MANDATORY OUTPUT FORMAT:**
```
STEP 1 - Individual Scores:
Clarity: X/35 (reason for score)
Stability: X/20 (reason for score)
Lighting: X/20 (reason for score)
Artifacts: X/15 (reason for score)
Occlusion: X/10 (reason for score)

STEP 2 - Base Score Calculation:
Base Score = X + X + X + X + X = X/100

STEP 3 - Bonus/Penalty Adjustments:
[List each bonus/penalty with reason and points]
Total Adjustment: +/-X points

STEP 4 - Final Results:
Final Score = X (Base) + X (Adjustment) = X/100
Quality Tier: [Premium/High/Standard/Below Standard]
Critical Issues: [List any issues that prevent premium quality classification]
Motion Blur Check: [Specifically note if eyes/mouth/teeth show any motion blur]
```

**IMPORTANT GRADING NOTES:**
- Be exceptionally strict with scoring - err on the side of lower scores
- Only award top scores (90+) for truly exceptional, near-perfect quality
- Consider that this is for premium training data - standards are higher than typical use
```

```
- Focus on details that would be critical for face restoration model performance
- Penalize any imperfections that could negatively impact training effectiveness
- DOUBLE-CHECK your arithmetic at each step to ensure accuracy

Please provide a complete evaluation following the exact format above, including all
    calculation steps.
```

# C  USE OF LARGE LANGUAGE MODELS

Large Language Models (LLMs) were used only for grammar checking and text polishing. All research ideas, methods, and analyses are solely by the authors.

