# OpenReview forum: "VividFace: High-Quality and Efficient One-Step Diffusion For Video Face Enhancement"
_ICLR.cc/2026/Conference — ICLR 2026 Conference Withdrawn Submission_

### Official Review · Reviewer_fJMu · 2025-10-29

**Soundness:** 3
**Presentation:** 3
**Contribution:** 3
**Rating:** 6
**Confidence:** 2

**Summary:**

The paper proposes VividFace, a one-step diffusion framework for video face enhancement, leveraging the pretrained WANX model. Key innovations include a joint latent-pixel face-focused training strategy, an MLLM-driven data curation pipeline, and a single-step flow matching paradigm. The method claims superior performance in perceptual quality, identity preservation, and temporal consistency, with significantly faster inference than existing approaches.

**Strengths:**

1.	Novelty: The work introduces the first one-step diffusion framework tailored for video face enhancement, combining flow matching with face-specific optimization. The joint latent-pixel training and MLLM-based data filtering are innovative contributions.

2.	Comprehensive Evaluation: Extensive experiments on synthetic (VFHQ-test) and real-world (RFV-LQ) benchmarks demonstrate state-of-the-art results across multiple metrics (e.g., PSNR, LPIPS, IDS).

3.	Efficiency: The one-step inference achieves a 12× speedup over SVFR and 2.7× over SeedVR2-3B, making it practical for real-time applications.

**Weaknesses:**

Reproducibility:

1）The reliance on the proprietary WANX model (architecture/training details undisclosed) may hinder reproducibility.

2）The MLLM-Face90 dataset is not publicly released; details on its curation (e.g., MLLM prompts, filtering thresholds) are insufficient.

Baseline Comparisons: Some baselines (e.g., SVFR, SeedVR2) are compared using their default settings, but adaptations for fair comparison (e.g., same training data) are not discussed.

Limitations: The method is evaluated only on face-centric videos; generalization to non-facial video restoration is untested.

**Questions:**

Please refer to the paper weakness

---

### Official Review · Reviewer_g8YP · 2025-10-31

**Soundness:** 2
**Presentation:** 2
**Contribution:** 2
**Rating:** 2
**Confidence:** 5

**Summary:**

The authors focus on the problem of video face restoration and propose a one-step diffusion framework called VividFace for video face enhancement. Specifically, VividFace leverages a pretrained WANX video generation model and reformulates the traditional multi-step diffusion process into a single-step paradigm, significantly reducing inference time. The method includes a Joint Latent-Pixel Face-Focused Training strategy and an MLLM-driven data curation pipeline to enhance facial detail recovery and model generalization. The proposed method achieves better performance than existing methods in perceptual quality, identity preservation, and temporal stability.

**Strengths:**

1. The proposed one-step diffusion method achieves state-of-the-art visual quality at a remarkable speed (~12x faster than strong baselines), effectively addressing the critical efficiency bottleneck for practical applications.
2. The MLLM-driven data curation pipeline is valuable, and its effectiveness is empirically validated by the performance boost from training on the resulting MLLM-Face90 dataset.
3. The Joint Latent-Pixel Face-Focused Training is a well-designed strategy that proves crucial for enhancing fine-grained facial details while preserving global consistency.

**Weaknesses:**

1. Despite the one-step approach, the overall framework remains complex due to the integration of multiple components, such as the Joint Latent-Pixel Face-Focused Training strategy and the MLLM-driven data curation pipeline.

2. The MLLM filtering pipeline is not rigorously validated. There’s no comparison showing how MLLM-Face90 compares to human-curated subsets or alternative filtering strategies. This lack of validation makes it difficult to assess the true effectiveness and reliability of the MLLM-driven approach in selecting high-quality training data.

3.  While the paper uses a comprehensive set of metrics, the reliance on quantitative metrics alone might not fully capture the subjective quality of the enhanced videos. Human perception studies or user evaluations could provide additional insights into the practical effectiveness of the method.

**Questions:**

1. The method builds upon the powerful pre-trained video generation model. To better understand the specific contributions of the proposed fine-tuning strategy, could the authors report the baseline performance of the original WANX model on this task? This would help disentangle the gains from the fine-tuning process versus the inherent capabilities of the powerful base model.

2. The MLLM-driven data curation is a novel approach. Given that MLLMs can be prone to hallucination and rating instability, could the authors elaborate on the potential impact of these issues on the final dataset quality and any steps taken to mitigate them? Additionally, could you provide the rationale for preferring this MLLM-based method over more established and objective non-reference video quality assessment (NR-VQA) metrics for the filtering process?

3. To further strengthen the real-world evaluation, have the authors considered comparing against leading Face Image Restoration (FIR) methods applied frame-by-frame? Similarly, to provide a more direct measure of temporal consistency, would it be possible to report results using the VIDD metric introduced by the work [a]?

4. The work [a] introduced two widely recognized benchmarks for real-world face video restoration. To more comprehensively demonstrate the method's robustness and generalization, have the authors considered evaluating VividFace on these two benchmarks?

[a] Towards real-world video face restoration: A new benchmark. CVPR, 2024

---

### Official Review · Reviewer_2PT2 · 2025-10-31

**Soundness:** 3
**Presentation:** 3
**Contribution:** 2
**Rating:** 4
**Confidence:** 4

**Summary:**

This paper introduces an efficient and effective face enhancement method.

The proposed model is rectified for fast inference and incorporates several training strategies to improve performance. Additionally, the authors curate a new dataset specifically designed for this task.

**Strengths:**

1. The method adopts a one-step design, enabling efficient and fast face enhancement.
2.	Based on both qualitative and quantitative evaluations, the approach achieves state-of-the-art performance within this domain.
3.	The data creation and selection pipeline and the newly constructed dataset appear effective, as supported by the ablation study.
4.	The proposed two-stage and mask-based training strategies demonstrate measurable improvements in performance.

**Weaknesses:**

1.	The masking techniques and rectified model training strategies used in this work are not entirely novel and have been employed in prior studies.
2.	The primary performance gain seems to stem from using a larger pre-trained backbone and a better-curated dataset, rather than from a fundamentally new algorithmic contribution. The authors should better clarify the technical novelty and distinct contributions of this paper.
3.	The ablation results show relatively small metric improvements; including qualitative examples would make the analysis more convincing.

**Questions:**

1.	While the reported performance is strong, the technical novelty requires clearer justification—what aspects are truly new beyond model scaling and data curation? This is my main concern.
2.	Will the proposed dataset be publicly released to support further research in this area?
3.	How dependent is the method on the pre-trained model? Please clarify the parameter count and scale differences compared to baseline methods.

**Details Of Ethics Concerns:**

The paper involves the use of human video data, yet it does not discuss privacy considerations or how personally identifiable information is handled. The authors should include a section addressing data privacy, consent, and ethical implications of using such data. Additionally, potential ethical concerns—such as the misuse of this method—should be discussed to ensure responsible research practices.

---

### Official Review · Reviewer_YuHS · 2025-11-01

**Soundness:** 3
**Presentation:** 2
**Contribution:** 2
**Rating:** 4
**Confidence:** 4

**Summary:**

This paper introduces VividFace, a novel and efficient one-step diffusion framework designed to address the core challenges in Video Face Enhancement (VFE). The authors identify key limitations in existing methods: difficulties in balancing texture detail with temporal consistency, poor generalization due to lack of high-quality data, and low inference efficiency. VividFace tackles these by building on a pretrained video model to leverage strong spatiotemporal priors, reformulating the diffusion process into a single-step flow matching paradigm for significant speed gains. It further proposes a joint latent-pixel training strategy with stochastic switching for focused facial optimization, and an MLLM-driven pipeline to curate high-quality training data. Extensive experiments validate that VividFace achieves state-of-the-art performance in perceptual quality, identity preservation, and temporal stability.

**Strengths:**

+ Exceptional Efficiency: A one-step diffusion framework enabling real-time, high-quality video face enhancement with a 12× speed boost.
+ Precise Facial Detailing: A joint latent-pixel training strategy that focuses restoration on key facial regions for superior visual fidelity.
+ High-Quality Data Curation: An automated MLLM-driven pipeline for reliable training data, enhancing model performance with authentic facial details.
+ The paper is easy to follow.

**Weaknesses:**

- This proposed framework is built upon the pretrained WANX video generation model. It seems that this paper used a better backbone or parameters to facilitate the learning of the model. However, the authors do not explore the effect of thie pretrained model. It seems that the comparisions of the proposed model with those without the pretrained model are unfair.
-  Through a single-step flow matching paradigm,  the spatiotemporal priors enabls direct mapping from degraded inputs to high-quality outputs with significantly reduced inference time.  I think this contribution is limited, as the spatiotemporal priors and the folow matching mechanism are common tecniques in the restorations of video faces. In the ablation studies, there are also no corresponding explorations.
- For the  JointLatent-Pixel Face-Focused Training strategy, it employs stochastic switching between facial region optimization and global reconstruction. The authors should analyse the fixing optimization strategy.
- The selection of the training data is also a common tecnique in the computer vision. The authors should explore the proposed data selection mechanism with other related techniques.
- There are no intrinsic connection and theoretical guarantee for the proporsed techniques  in this paper.
- The experimental datasets are simple, it seems that there is only on face in a frame. The authors should explore more complicated scenarios. For example, there are more faces in a frame, the faces are small, etc.

**Questions:**

Please see the weaknesses.

---

### Note · Authors · 2025-11-12

I have read and agree with the venue's withdrawal policy on behalf of myself and my co-authors.